# Recent Advances of Pluripotent Stem Cell-Derived Cardiomyocytes for Regenerative Medicine

**DOI:** 10.3390/metabo15110735

**Published:** 2025-11-11

**Authors:** Farag M. Ibrahim, Ahmed Atef, Mostafa M. Mostafa, Mohammed A. Sayed

**Affiliations:** 1Zoology Department, Faculty of Science, Benha University, Banha 13511, Egypt; farag199530@fsc.bu.edu.eg; 2Faculty of Medicine, Modern University for Technology and Information, Cairo 4411601, Egypt; ahmed16008@stemegypt.edu.eg; 3Department of Molecular and Cellular Physiology, Stritch School of Medicine, Loyola University, Chicago, IL 60153, USA; mmostafa@luc.edu; 4Department of Pharmacology, Debusk College of Osteopathic Medicine, Lincoln Memorial University, Harrogate, TN 37752, USA

**Keywords:** pluripotent stem cells, induced pluripotent stem cells, cardiomyocyte maturation, cardiac regeneration, heart failure, tissue engineering, cell therapy, cardiovascular disease

## Abstract

Cardiac muscle has limited proliferative potential; therefore, loss of cardiomyocytes is irreversible and can cause or exacerbate heart failure. Although both pharmacological and non-pharmacological therapies are available, these interventions act primarily on surviving myocardium to manage symptoms and reduce—rather than reverse—adverse remodeling. The only curative option for end-stage heart failure remains heart transplantation; however, its clinical use is severely constrained by the shortage of donor organs. Consequently, regenerative therapies have gained increasing attention as potential novel treatments. Among these, cardiomyocytes derived from patient-specific pluripotent stem cells (PSCs) represent a particularly promising experimental platform for cardiac regeneration. To evaluate the potential of PSCs for cardiac repair through both in vivo and in vitro approaches, we (1) examined the hallmarks of cardiomyocyte maturation and the regulatory systems that coordinate these processes, (2) reviewed recent advances in maturation protocols and derivation techniques, (3) discussed how the cellular microenvironment enhances maturation and function, and (4) identified current barriers to clinical translation. Importantly, we integrated developmental biology with protocol design to provide a mechanistic foundation for PSC-based regeneration. Specifically, insights from cardiac development—such as signaling pathways governing proliferation, alignment, and excitation-contraction coupling—were explicitly linked to the refinement of PSC differentiation and maturation protocols. This developmental perspective allows us to bridge pathology and stem-cell methodology, explaining how disruptions in native cardiac maturation can inform strategies to produce functionally mature PSC-derived cardiomyocytes. Finally, we assessed the clinical prospects of PSC-derived cardiomyocytes, highlighting both the most recent advances and the persistent translational challenges that must be addressed before widespread therapeutic use.

## 1. Introduction

Cardiovascular diseases are considered the second leading cause of death in the USA, accounting for more than 900,000 deaths per year [1]. This umbrella term encompasses various diseases, such as coronary heart disease (CHD), cerebrovascular disease, peripheral arterial disease, rheumatic heart disease, deep vein thrombosis, and pulmonary embolism. During cardiovascular diseases, Cardiomyocytes are irreversibly damaged. Adult cardiomyocytes (CMs) have limited regenerative ability, leading to a cascade of events that cause severe heart failure and, ultimately, the loss of CMs and potential scarring [2,3]. Pharmacological interventions that may delay or reverse the process of cardiac remodeling, such as angiotensin-converting enzyme (ACE) inhibitors, angiotensin receptor–neprilysin inhibitors, β-blockers, and mineralocorticoid receptor antagonists, have been effective in reducing mortality rates in individuals with heart failure [4,5,6,7]. Unlike the cardioprotective treatments that focus on modifying the remodeling process in the failing heart, only limited options exist for treating end-stage heart failure. While these patients may benefit from mechanical support treatments, including left ventricular assist devices (LVADs) and cardiac resynchronization therapy, heart transplantation remains the only existing remedy for regenerating an impaired heart. However, the scarcity of available donor organs remains challenging [8,9,10].

While the main issue with cardiovascular diseases (CVDs) is irreversible tissue damage, stem cell therapy, mainly using pluripotent stem cells (PSCs), is a promising area of research that aims to address heart damage by regenerating heart tissue and improving heart function. PSCs possess the capacity for unlimited proliferation and the capability to develop into cells originating from all three germ layers. These capabilities make them an appealing option for cell therapies targeting a range of illnesses and injuries. Two types of human PSCs (hPSCs) are being explored for clinical use: embryonic stem cells (hESCs) and induced pluripotent stem cells (iPSCs). hESCs were first reported by James Thomson’s group in 1998 [11], seventeen years after the generation of mouse ESCs. This prolonged time lag between the creation of mouse and human ESCs was due to substantial differences in morphology and culture conditions [11], as described later. Consequently, the potential of hESCs has been explored in cell therapies for various diseases [12].

Since the identification of hESCs and iPSCs (collectively referred to as hPSCs), researchers have been working on creating efficient and reliable differentiation processes to direct these cells toward a cardiomyocyte fate for in vitro and in vivo applications [11,12,13]. Multiple differentiation methods have been reported for producing functional CMs from hESCs. Recent progress in iPSC technology has shown that human iPSCs may be a suitable source for generating patient-specific, differentiated CMs in vitro, which share similarities with their hESC-derived counterparts but could provide a new avenue to personalized cardiovascular care [14,15]. (Figure 1) illustrates some of the processes, challenges, and outcomes associated with cardiomyocyte transplantation, starting from induced pluripotent stem cells (iPSCs) and progressing through immune rejection, fibrosis, arrhythmias, and cell death.

Due to their cardiac phenotype and functional properties, CMs derived from PSCs offer a more physiologically and clinically relevant human cell model. While directed differentiation has been used to generate CMs from PSCs in vitro, which provides an excellent platform for studying cell fate specification mechanisms [16,17], practical applications of PSC-based therapies for patients have raised concerns about low engraftment rates and the potential risk of tumorigenesis [16].

The most promising application for using hPSCs in treating cardiomyopathies is, perhaps, the recent study published by Jebran et al. in *Nature* [18], where the authors succeeded in using engineered heart muscle (EHM) allografts in treating advanced heart failure in both macaques and one human patient, with no evidence of arrhythmia or tumor growth.

This narrative review discusses the potential utilization of pluripotent stem cells (PSCs) in vivo and in vitro to generate cardiomyocyte cells to replace irreversibly damaged cardiomyocytes during cardiovascular diseases. We begin by offering insights into cardiomyocyte origin, plasticity, and differentiation from pluripotent stem cells; then, we explore the recent advances and obstacles accompanied by the potential utilization of pluripotent stem cells (PSCs) in generating cardiomyocyte cells.

## 2. Insights into the Cardiac Origin and Its Pathogenesis

The growing heart of the human fetus is the first full organ to develop [19]; cardiac abnormalities contribute significantly to embryonic death [20]. Heart progenitor cells are discovered in an anterior mesoderm region near the endoderm shortly after gastrulation [21]. The three germ layers, ectoderm, endoderm, and mesoderm, are defined early in human development. Most cardiac cells develop from BRY+ eomesodermin (EOMES)+ mesoderm, while MESP1+ cardiac mesoderm differentiates and divides into two heart zones, generating the cardiac crescent [22]. These cardiac fields create loop tubes to produce the heart’s four chambers: the right ventricle, right atrium, left ventricle, and left atrium [23,24]. The endocardium, mainly composed of specialized endothelial cells; the myocardium, which comprises CMs; and the epicardium, which comprises epicardial cells, are the three separate layers of a wholly formed heart wall.

Cardiomyopathy is a complex heart condition linked to electrical and/or mechanical malfunction. The walls of the heart chambers stretch, thicken, or stiffen during cardiomyopathy, which impairs the heart’s capacity to pump and transport blood to the body’s organs. It is divided into four primary categories and can be brought on by both environmental and genetic factors: HCM, or hypertrophic cardiomyopathy [25], dilated cardiomyopathy (DCM) [25], restricted cardiomyopathy (RCM), and arrhythmogenic right ventricular cardiomyopathy [26].

Importantly, the capacity to produce induced pluripotent stem cell-derived cardiomyocytes (hiPSC-CMs) specific to each patient has produced potent preclinical models for researching the distinct disease mechanisms of these subtypes, particularly the genetic forms (e.g., HCM and DCM), allowing for more individualized cardiotoxicity screening and drug development initiatives [25,26]. Dilated cardiomyopathy (DCM) is a clinical diagnosis characterized by left ventricular or biventricular dilation and impaired contraction that is not explained by abnormal loading conditions (for example, hypertension and valvular heart disease) or coronary artery disease. Mutations in several genes can cause DCM, including genes encoding structural components of the sarcomere and desmosome. Nongenetic forms of DCM can result from different etiologies, including inflammation of the myocardium due to an infection (mostly viral); exposure to drugs, toxins, or allergens; and systemic endocrine or autoimmune diseases [25]. DCM is therefore a crucial target for PSC-based strategies; PSC-CM transplantation is a promising clinical strategy to remuscularize the failing myocardium in end-stage DCM, while patient-derived hiPSC-CMs function as preclinical models to study genetic and non-genetic disease mechanisms and screen therapies [25].

Hypertrophic cardiomyopathy (HCM) is a genetic disorder that is characterized by left ventricular hypertrophy unexplained by secondary causes and a non-dilated left ventricle with preserved or increased ejection fraction. It is commonly asymmetric, with the most severe hypertrophy involving the basal interventricular septum [26]. Similar to DCM, HCM patients’ hiPSC-CMs are very useful for simulating the distinct cellular pathophysiology and evaluating the effectiveness of medications, particularly for new sarcomeric modulators. The other two forms (restrictive cardiomyopathy and arrhythmogenic right ventricular cardiomyopathy) are rare, and their diagnoses require a high index of suspicion [27]. These conditions often result in progressive heart failure or cardiovascular death. Heart failure is a significant global health concern, leading to high rates of mortality and morbidity worldwide [28]. While current treatments, such as pharmacological, surgical interventions, and devices, can help delay the progression of heart failure, they do not address the root cause of the disease [29]. In severe cases, heart transplantation or the use of artificial hearts remains the most effective treatment option. However, the limited availability of donor hearts and the high costs associated with transplantation procedures pose challenges in providing this treatment to all in need [30]. It is important for individuals with a family history of cardiomyopathy or related symptoms to seek medical evaluation and genetic counseling to understand their risk factors and potential preventive measures. Early detection and management can significantly improve outcomes for individuals with cardiomyopathy [31].

Another form of pathological heart conditions is myocardial infarction (MI). MI occurs due to decreased or complete cessation of blood flow to a part of the heart muscle, leading to cardiomyocyte death from lack of oxygen supply [1]. Since ischemic cardiomyopathy is essentially a disease of cell loss, as opposed to hereditary cardiomyopathies, it is the primary focus of clinical PSC-CM transplantation therapy, which aims to replace dead tissue and restore contractile function. Human cardiomyocytes have a limited ability for regeneration, and available medications do not provide workable methods for replacing lost cells following MI [32]. Clinical results for certain heart conditions may therefore be improved by the possible application of cardiac regenerative medicine, which permits cardiomyocyte replacement. Although there are still numerous obstacles in the way of their application, researchers have been looking at the possibility of using stem cell therapy to treat MI. The survival and cell fate of transplanted cells can be impacted by a hostile ischemic environment, fibrotic scarring, immunological reactions, inadequate vascularization, and a shortage of donors [33,34,35].

## 3. The Use of Pluripotent Stem Cells in Generating Cardiomyocytes and Cardiac Regenerative Medicine

The PSCs, encompassing both ESCs and iPSCs, hold significant promise in regenerative medicine for treating heart diseases because of their nearly limitless ability to generate cardiomyocytes. Numerous preclinical studies have highlighted the functional advantages following the transplantation of PSC-derived cardiomyocytes (PSC-CMs) [36].

Experiments using PSCs have shown that they may be used to simulate human cardiac development, demonstrating that cardiovascular cells could be generated in vitro, even in poorly defined conditions such as fetal bovine serum (FBS) [37]. These studies showed that the differentiation of PSCs underwent developmental processes resembling early embryonic heart development. The first report of cardiac myocyte production from PSCs employed embryoid bodies in serum-containing media [38] and succeeded with only 5% to 10% efficiency. Several groups have since developed methods for increasing efficiency. Mummery et al. showed that the inclusion of mouse endodermal cells (END2) in the culture enhances the efficacy of differentiation [39]. Cardiac myocyte generation was achieved efficiently by employing cytokines, namely BMP4 (bone morphogenetic protein 4) and Activin A, in both 2-dimensional monolayer and embryoid body-based differentiation systems [38,39]. Using this strategy, researchers observed that activinA/Nodal, Wnt, and BMP, which are known to influence primitive streak (PS) formation and induction of mesoderm [40], assisted in the efficient synthesis of cardiac populations of hPSC [40,41,42].

The use of small compounds to modulate Wnt/-catenin and BMP signaling in hPSCs to control cardio-myogenesis is promising. Human iPSCs were treated with BMP-4 early on, followed by small molecule Wnt inhibitors, resulting in a considerable elevation in the rate of cardiomyocyte production compared to earlier differentiation approaches [43]. The findings provided information on the temporal mechanism by which Wnt activation regulates human induced pluripotent stem cell-derived cardiomyocytes (hiPSC-CM) development and guidelines for consistently creating hiPSC-CMs in cardiovascular research [43]. Moreover, Sante et al. showed that MAO-A-mediated ROS generation is necessary to activate AKT and WNT signaling pathways during cardiac lineage commitment and to differentiate fully functional human cardiomyocytes [44]. Furthermore, early treatment with a Wnt production inhibitor inhibited anti-cardiac mesoderm gene expression and increased cardiomyocyte formation by up to 80%, despite low cell density cultures producing more anti-cardiac mesoderm genes [45]. According to these studies, the auto/paracrine pathways’ reduction of Wnt signaling early in the induction phase is the main effect of cell density on cardiomyocyte formation [45].

Furthermore, nicotinamide increases the formation of CMs from mesoderm progenitor cells while suppressing the emergence of other cell types [46]. This occurs by inhibiting p38 MAP kinase. In addition, nicotinamide, which inhibits a Rho-associated protein kinase (ROCK), improves cardiomyocyte lifespan [46]. Burridge et al. presented a three-step differentiation system to produce cardiac myocytes using only chemically defined factors without serum [47]. Salerno et al. hypothesized that increased S-nitrosylation increases the differentiation and maturation of iPSC-derived CMs [48]. This hypothesis depended on the fact that nitric oxide signaling promotes the development and differentiation of CM in the absence of S-nitroso glutathione reductase (GSNOR), a denitrosylase that controls protein S-nitrosylation.

However, not all iPSC lines efficiently develop into CMs. Aguado et al. showed that compared to iPSCs with reduced expression of TRF1 and comparatively short telomeres, iPSCs that are telomerase competent and have a high level of the shelterin-complex protein TRF1 expression of the sheltering complex protein (iPSC^high^T) differentiate more quickly and effectively into CM [49]. The production of CMs from iPSC^high^T is further increased by ascorbic acid, an amplifier of cardiomyocyte differentiation, whereas iPSC’s cardiomyogenic capacity is not restored. Interestingly, iPSCs^low^T differentiates poorly in the mesoderm and endoderm lineages but very well in the ectoderm lineage, demonstrating that cell destiny may be controlled by in vitro selection of iPSCs with varied telomere levels [49]. Moreover, Vervliet et al. discovered that the differentiation of hiPSCs to cardiomyocytes was delayed following BCL2 KO [50]. This was unrelated to the canonical anti-apoptotic role of Bcl-2. The delay decreased the expression and activity of the Ca2+ toolbox in the cardiomyocytes [50]. In addition, the knockout of Bcl-2 reduced the expression of c-Myc and its nuclear localization during the first stage of cardiac differentiation [50]. This partially explains the observed delay in the process of cardiac differentiation.

Because several studies have shown that high cell density culture improves cardiomyocyte development, it has been proposed that high cell density culture might promote heart cell differentiation. For instance, cells are plated at 1–4105 cells/cm^2^ before cardiac induction and given several days to attain confluence [51,52,53,54]. However, the intricate connections between cells in a dense environment make understanding the functions of cell signals challenging. Le et al. calculated the minimal cell density that can induce differentiation to help in cell-cell interaction investigations [55]. The authors also used a basic medium deficient in essential components to maintain the pluripotency of hiPSC, such as primary fibroblast growth factor, activin A, and insulin. Cell survival was enhanced by adding insulin and a Rho-associated protein kinase inhibitor to the starting medium. However, cardiac markers were only made at the mRNA level but not at the protein level when the primary fibroblast growth factor was added. The cardiac troponin T protein, linked to cell contraction, was produced by 10% of the cells after additional changes were made to the growth conditions. The outcomes will help identify the cell signals necessary to develop CMs [55]. Studying the fundamental components driving cardiomyocyte growth at a low initial cell density helped researchers understand the effect of cell density. Auto/paracrine factors, rather than cell-cell contact signals, played a significant role in cardiomyocyte development in HiPSC coculture at low and high cell densities [45].

Cell death is a detrimental factor during in vitro cardiomyocyte differentiation. Aalders et al. [56] concluded that ferroptosis was the primary cell death modality during the first 48 h of the cardiac differentiation procedure. Inhibiting ferroptosis during cardiomyocyte differentiation increases both the robustness and the efficiency. Kadota et al. [57] showed that, after 3 months of transplantation of hiPSC-CM into neonatal rat hearts, engrafted hiPSC-CMs developed partially matured myofibrils, with increases in cell size and sarcomere length. However, these cells remained smaller compared to the host rat cardiomyocytes, indicating that while some maturation occurred, it was not complete. In contrast, hiPSC-CMs transplanted into adult rat hearts exhibited greater maturation than those in neonatal hearts. The grafts in adult hearts were larger, and the cells showed more advanced structural characteristics compared to those in neonatal hearts. Their study concluded that the growing neonatal rat heart does not serve as an effective bioreactor for enhancing the maturation of hiPSC-CMs. Instead, the adult rat heart provided a more conducive environment for the maturation of these cells, suggesting that the developmental stage of the host heart significantly influences the maturation process of engrafted hiPSC-CMs. Overall, while some maturation of hiPSC-CMs occurs in neonatal hearts, the adult heart environment promotes a higher degree of maturation, highlighting the importance of the host’s developmental stage in stem cell therapy outcomes, Table 1 [57,58].

## 4. Cardiomyocyte Maturation Methods to Produce Functional Cardiomyocytes

The term “cardiomyocyte maturation” denotes changes in metabolism, gene expression, function, and cell structure throughout the shift from fetal to adult CMs [58]. Although recent advancements permit the successful conversion of PSCs into CMs, These PSC-CMs have immature characteristics comparable to fetal CMs [61,62], and resemble cardiac myocytes seen in the heart, composed of nodal, ventricular, atrial-like, and with varying electrical properties. Cell therapy applications require the ability to produce mature cell types that match the illness’s target. For example, cell-based treatments for post-MI patients will need ventricular CMs to have highly enriched populations [63]. At the same time, atrial cells will be necessary to mimic disorders such as atrial fibrillation [64].

The sinus node organizes the exact sequence of electrical events that sustain the regular cardiac rhythm [65]. Retinoic acid and BMP signaling are altered during cardiac myocyte maturation, which improves nodal-like cell generation efficiency [66]. Schweizer et al. co-cultured hiPSCs with visceral endoderm-like cells in serum-free media to grow them into spontaneously beating clusters [67]. The cells developed a pacemaker-like phenotype after being cultivated in a specific FBS-enriched cell medium [68]. Overexpression of the transcription factors TBX18, TBX3, and HCN2 increases the production of pacemaker-like cells in hiPSC, according to Gorabi [69,70,71].

It has been shown that retinoic acid (RA) treatment increased the differentiation of atrial cardiac myocytes [72]. Several studies have shown that activating RA signaling leads to forming populations enriched with atrial CMs [73,74]. Moreover, Gao et al. showed that ascorbic acid induces MLC2v protein expression and promotes ventricular-like cardiomyocyte subtype in hiPSCs-CMs [75]. Furthermore, Dark et al. showed that it is possible to quickly generate nearly identical human left ventricle cardiomyocytes from human pluripotent stem cells [76]. These cells demonstrate a higher level of maturity than other cardiomyocyte cultures of the same age [76]. These cells are homogeneous and mature, making them a viable model for studying the development and pathology of the left ventricle. Additionally, they should allow for more accurate cardiotoxicity screening [76].

Building on these accomplishments, the next stage is understanding and regulating cell maturation. Most procedures create cells during the embryonic or early fetal phases, often shortly after organogenesis [77]. As a result, the produced cells lack many adult cellular features useful for replacing diseased cells [78]. Several methods were developed to improve the maturation outcomes of hiPSCs-CMs, which are summarized in the following section.

## 5. Strategies to Improve the Outcomes of Hipscs-CM Maturation

Cardiomyocyte maturation in vitro requires recapitulating the complex biochemical and physical environment of the postnatal heart. In vivo, maturing cardiomyocytes undergo a dramatic metabolic transition—shifting from predominantly glycolytic energy production in the fetal period to robust mitochondrial fatty acid oxidation in the adult heart [58,79].

Consequently, effective maturation protocols combine multiple cues: genetic and epigenetic modifiers (e.g., microRNAs), hormones (e.g., thyroid hormone and glucocorticoids), metabolic stimuli (substrate and oxygen conditions), and biomechanical conditioning (electrical pacing, stretch, and 3D tissue context). Each of these factors drives aspects of structural, electrophysiological, and metabolic maturation (for example, changes in cell size, sarcomere alignment, ion channel expression, and excitation–contraction coupling) that distinguish adult cardiomyocytes from their fetal-like PSC-derived counterparts (see Table 2 for factors enhancing structural, electrophysiological, and metabolic maturation of pluripotent stem cell–derived cardiomyocytes).

However, recent metabolomic profiling of PSC-derived cardiomyocytes has begun to map these energy-state differences in detail [80,81]. Such studies find that mature PSC-CM cultures upregulate fatty acid oxidation and lipid-processing pathways, and even specific metabolite signatures (e.g., increased glycerophosphocholine, decreased phosphocholine) that mirror neonatal heart development. In contrast, standard PSC-CM media (high glucose, low lipid) tend to “lock” cells in a glycolytic, immature state [80,81]. By systematically comparing the metabolome of PSC-CMs at various stages to that of adult myocardium, researchers can identify which substrates and pathways are deficient. This data-driven insight then guides culture optimization: for example, lowering glucose, adding fatty acids or metabolic cofactors, or applying hypoxia/HIF1 inhibitors and PPAR activators to boost oxidative metabolism. Indeed, supplementing cultures with fatty acid mixtures (palmitate, oleate, linoleate) or inhibiting glycolytic signaling has been shown to increase cell size, mitochondrial function, and contractile force in PSC-CMs. Overall, integrating metabolomic profiling into differentiation protocols provides biomarkers to track maturation and reveals how to tweak media composition and timing to expedite the metabolic switch that is crucial for adult-like cardiomyocyte function.

**Table 2 metabolites-15-00735-t002:** Summary of factors enhancing structural, electrophysiological, and metabolic maturation of pluripotent stem cell–derived cardiomyocytes.

Category	Factor/Treatment	Maturation Markers	Mechanism/Pathway/Observation	Source/Study
microRNAs	miR-1	Shortening action potential duration (APD), hyperpolarizing resting membrane potential, increasing upstroke velocity and contractile gene expression, suppressing automaticity, and promoting electrophysiological maturation and did not bias the yield.	Context-dependent regulation of cardiac development	[82]
	miR-499	Shortening APD, hyperpolarizing resting membrane potential, increasing β-MHC and Cx43 expression, reducing spontaneous beating rate, and promoting ventricular determination, yielding 72%.	Context-dependent cardiac differentiation effect	[82]
	Let-7	Enhancing cell size, sarcomere length, force of contraction, and respiratory capacity.	Inhibits the PI3K/AKT pathway and promotes fatty acid metabolism	[83]
Hormonal Regulation	Triiodothyronine (T3)	Adult-like structural, electrophysiological, and metabolic features, including organized sarcomeres, improved ion channel function, and a shift toward oxidative metabolism.	Essential for postnatal heart function; increases after birth	[84]
	T3 therapy (1 week)	T3 increased cardiomyocyte size, anisotropy, and sarcomere length	Upregulates CDK inhibitor p21 and reduces DNA synthesis	[85]
	Thyroid + Glucocorticoid hormones	Inducing T-tubule formation, enhancing Ca-induced Ca release coupling, and promoting more adult-like excitation–contraction function.	Synergistic hormonal effect enhancing hiPSC-CM functional maturation	[86]
Metabolism and Hormonal Interventions	Fatty acid supplementation (palmitate, oleate, linoleate)	increase human cardiomyocyte hypertrophy, force generation, calcium dynamics, action potential upstroke velocity, and oxidative capacity with over 95% positive cardiac troponin T.	Promotes metabolic shift toward fatty acid oxidation	[87]
	Sequential medium (lactate+/glucose–→fatty acids+/glucose–)	Inducing more adult-like energy metabolism (higher mitochondrial oxidative function, greater fatty acid use), enhanced structural organization, and upregulation of mature cardiac genes with 98% purity.	Enhances mitochondrial activity (OCR, ATP production, peak/baseline respiration, spare respiratory capacity)	[88]
	High glucose levels	Inhibiting HIF-1α or its downstream target LDHA in hPSC-derived cardiomyocytes shifts metabolism from glycolysis toward oxidative phosphorylation, increases mitochondrial content and ATP levels, and enhances structural and functional maturation	Activate HIF1 → upregulate glycolysis and LDH, inhibit oxidative phosphorylation	[89,90,91,92]
	HIF1 inhibition ± PPAR agonist + IGF1 + Dexamethasone + T3	Increased FAO and improved mitochondrial maturation	HIF-1α inhibition with PPARα activation and the postnatal factors	[93,94]
Mechanical and Electrical factors	Mechanical conditioning in 3D cardiac tissue	Adult-like heart tissue structure, overexpression of ITPR3, CAV3, MYH7, RYR2, and KCNH2 and down-regulation of HCN4 and MYH6. Aligned sarcomeres, high mitochondrial density (30%), abundant T-tubules, adult-like calcium handling, robust calcium-induced response, positive force-frequency link.	Imitates the heart’s rising contractile demand after birth. High-intensity training regimen including electrical stimulation.	[95,96,97,98,99]
	Electrical stimulation	Increased maturation of iPSC-CMs. Adult-like heart tissue structure.	Applied alongside mechanical stress (2–6 Hz for two weeks, then 2 Hz for an additional week).	[99]
	Electrical maturation (Development)	Increased IK1 and Ito currents. Increased calcium storage in the RYR-mediated sarcoplasmic reticulum. Transverse tubules appear after birth.	Resting membrane potential becomes more negative due to increased IK1 channel expression. Crucial for calcium control and excitation-contraction coupling.	[100]
Three-dimensional culture system	3D cultures	Adult phenotypic maturity: defined electrophysiological properties, structured sarcomere structure (I bands, H zones, M lines), distinctive gene expression. Enhanced advances in calcium management, force production, and electronic and mechanical coupling.	Provides a more physiologically appropriate in vitro paradigm mimicking a small organ. Cells form signal-transmission connections. Dynamic (rocking) platform for hCMP creation.	[101,102,103,104,105]
Extracellular matrix systems	Dynamic ECM application (e.g., Matrigel)	Significant cardiac differentiation, epithelial-mesenchymal transition, complementary growth factor signaling.	ECM (containing Activin A, essential fibroblast growth factor, and BMP4) used as a substrate.	[106]
	Natural cardiac ECM and 3D cultures	Significantly enhanced maturation compared to 2D. Increased expression of calcium-handling genes (HCN4, Junctin, SERCA2a, NCX1, Triadin, CaV1.2, CASQ2).	3D scaffold made of ECM. Adding proteins (laminin-521 + laminin-221 or Synthemax) can enhance differentiation.	[107,108]
Cell sheet technology	Cell sheet technology	Development of cardiac tissue sheets that can be transplanted.	Extracts confluent cells grown on a temperature-responsive culture plate as a complete sheet. Can be differentiated into CMs and vascular cells simultaneously.	[109,110]
Microtissue platform	Spheroids/Micromasses (cardio-spheres)	Increased proportion of CMs (80–100%) compared to 2D (10–40%). Multiple maturation indications including increased expression of contractile machinery proteins and gap junction, fibril alignment and ultrastructure.	Simplest 3D method relying on cell self-assembly. Re-aggregation techniques (dangling droplets, low adhesion/concave plates, rotating cultures). Cyclic uniaxial mechanical stretching and electrical modeling applied to spheroids.	[111,112,113,114]
Heart-on-chip	Heart-on-chip culture system	Electrophysiologically separate atrial and ventricular tissues with chamber-specific pharmacological responses and gene expression. Improved intracellular structure, enhanced consistency, and function of the tissue.	Simulates blood circulation using microfluidic technology. Controls culture conditions (oxygenation, pH, shear stress, temperature, electrical/mechanical stimulation). Non-invasive online recording of contractile dynamics, force, Ca^2+^ transients, conduction velocity, and action potential.	[115,116,117,118,119,120,121,122]
3D bioprinting	3D bioprinting	Excellent printing fidelity and resolution. High orientation index of HUVECs (vascular cells) facilitating integration with the host’s vasculature. Significant improvement in cardiac function, reduced infarct size and apoptosis, increased vascular and arteriole density.	Enables simultaneous insertion of cells and biomaterials. Uses heterogeneous, multicellular structures (CMs, HUVECs/SMCs, ECs). Multiphoton-excited 3D printing to fabricate an hCMP that closely resembles the natural ECM scaffold.	[123,124,125]

Despite these various strategies, many limitations still exist in iPSC-CM maturation. For example, iPSC-CMs do not fully achieve adult cardiomyocyte maturity, particularly in electrophysiological properties, contractile function, and metabolic profiles. In addition, these cultures could be heterogeneous, including cells at different stages of maturation, which can cause problems with scalability and reproducibility and complicate their use in cells and therapy. Addressing these limitations requires research efforts to combine/refine maturation techniques, boost our understanding of maturation biology, and develop more efficient, cost-effective methods for producing fully mature iPSC-CMs.

## 6. Methods for In-Vivo Transplantation of PSC-CMs

PSC-CMs can be implanted in two main ways: as dispersed cells or as tissue-modified cells. Each has benefits and drawbacks [126,127,128,129]. In the scattered approach, the cell suspension is simple to prepare, cryopreserve, and inject into the inner muscle layer of the heart, but suspension cells leak from the injection site. They might not survive in the host heart, particularly early after transplantation [126,127,128,129]. Tissue-engineered cells, on the other hand, easily connect to the heart’s outer layer and may be retained and survive, resulting in decreased cell loss soon after transplantation [130,131,132]. Electrical integration of the host-graft is the most visible difference between the two procedures.

According to their findings, Gerbin et al. showed that unlike cardiac patches, which do not integrate electrically with the host, microtissue particles and scattered cells do [133]. The researchers tested the two methods by implanting Ca^2+^ indicator GCaMP-tagged PSC-CMs into the hearts of rats. The electrical connection has been observed in guinea pig hearts when engineered PSC-CM patches were engrafted on the epicardial surface, despite significant scar tissue between the host and graft. Therefore, dispersed cells or microtissue pieces can come into direct contact with each other to create gap junctions between the graft and the host [131]. Due to the direct connection between the graft and the host, dispersed cells might be a source of arrhythmias if infected cells have different electrical properties. Most studies have employed PSC-CMs that are roughly 20 days old and have been transplanted into animal models. By contrasting the bioluminescent intensity of PSC-CMs in mouse-infarcted hearts, Funakoshi et al. discovered that 20 days is the best time-point for transplantation [134].

## 7. Preclinical Applications of PSC-CMs in Treating Cardiac Disease

The research conducted by Cheng et al. [135] revealed that the cotransplantation of human induced pluripotent stem cell-derived cardiomyocytes (hiPSC-CMs) and endothelial cells (hiPSC-ECs) significantly enhances cardiac function in both murine and non-human primate models following myocardial infarction (MI). This strategy improved graft vascularization and promoted the maturation of cardiomyocytes, facilitating better integration with the host heart tissue. The findings indicated that the combination of hiPSC-CMs and hiPSC-ECs led to a significant preservation or enhancement of cardiac function, as assessed by metrics such as fractional shortening and ejection fraction, in comparison to control groups that received either cell type alone or no cells at all. Additionally, the endothelial cells were shown to have trophic effects on the grafts, which contributed to the improved survival and functionality of the cardiomyocytes. This underscores the critical role of endothelial cells in supporting cardiomyocyte grafts after transplantation, suggesting that the combined use of hiPSC-CMs and hiPSC-ECs could be a promising approach for enhancing heart regeneration following ischemic injury [135].

In a porcine model of myocardial infarction, a study investigated the potential of thymosin 4 (Tb4) to enhance the effectiveness of transplanted human induced pluripotent stem cell-derived cardiomyocytes (hiPSC-CMs) in myocardial infarction [136]. The results highlight that combining hiPSC-CMs with Tb4 treatment improves hiPSC-CM engraftment, stimulates vasculogenesis, enhances the proliferation of cardiomyocytes and endothelial cells, improves left ventricular systolic function, and reduces infarct size. Notably, the transplantation of hiPSC-CMs in immunosuppressed pigs did not lead to an increase in ventricular arrhythmia incidence or carcinogenesis [136].

In another study, Zhao et al. investigated the potential of increasing cyclin D2 expression on the outcome of the hiPSC-CMs in myocardial infarction [137]. The authors found that increasing cyclin D2 expression resulted in six-to-eight-fold enhancement in engraftment outcome compared to wild-type cells [137]. Moreover, using engrafted hiPSC-CMs with higher levels of Cyclin D showed marked improvement in left ventricular function compared to regular hiPSC-CMs [137].

Another potential modulator that could boost hiPSC-CM engraftment is Angiopoietin-1 [138]. Ang-1 protects hiPSC-CMs (human induced pluripotent stem cell-derived cardiomyocytes) from hypoxia-induced damage by activating ERK1/2 and inhibiting Bax protein, reducing cell death. In a rat model of myocardial infarction (MI), Ang-1-transfected hiPSC-CMs significantly improved heart function and structure, reducing left ventricular (LV) dilation and increasing LV wall thickness. Ang-1 promotes the mitosis (cell division) of hiPSC-CMs, enhancing their proliferation both in vitro (in the lab) and in vivo (in living organisms). Transplanted Ang-1-hiPSC-CMs showed better engraftment and survival rates in the damaged heart tissue, with reduced apoptosis (cell death) and increased formation of new blood vessels (vasculogenesis). Combining Ang-1 with hiPSC-CMs offers a more effective treatment for MI by not only regenerating heart muscle but also improving blood vessel formation and overall heart repair. In summary, transient overexpression of Ang-1 in hiPSC-CMs enhances their potential for heart repair by protecting cells, promoting their growth, and improving heart function and structure after a heart attack. This combined approach shows promise for developing new treatments for myocardial infarction [138].

Studies in large animals have shown that administering cardiomyocytes derived from human embryonic stem cells to infarcted non-human primate hearts can enhance cardiac healing [128]. However, the effectiveness of therapeutically relevant doses of hiPSC-CMs in large animal models of myocardial damage remains uncertain because of the challenges associated with poor engraftment of cells administered intramyocardially [128].

Chong et al. [126] successfully produced hESC-CMs at a clinical scale, exceeding 1 billion cells per batch, and demonstrated that these cells could be cryopreserved while maintaining good viability. Moreover, when hESC-CMs were delivered intra-myocardially in a non-human primate model of myocardial ischemia-reperfusion, significant remuscularization of the infarcted heart was observed. The grafts were able to integrate with the host vasculature and showed electromechanical junctions with host myocytes within two weeks of engraftment. The hESC-CMs exhibited progressive maturation over a three-month period. They demonstrated regular calcium transients that were synchronized with the host’s electrocardiogram, indicating effective electromechanical coupling, unlike previous studies in small animal models, the engrafted primates experienced non-fatal ventricular arrhythmias. This highlights a potential complication that needs to be addressed in future applications of this therapy in humans. The grafts comprised an average of 40% of the infarct mass, indicating a substantial degree of remuscularization, which is critical for restoring heart function after myocardial infarction [126].

## 8. Remaining Obstacles to Clinical Application

### 8.1. Immune Rejection

Implanting human pluripotent stem cell-derived cardiomyocytes (PSC-CMs) can elicit immune responses, possibly resulting in rejection. Even when pre-screened and safety-validated HLA-homozygous iPSCs are used, the immune system, in some instances, may still mount a response to eliminate the introduced cells. It is important to note that rejection can occur not only due to HLA mismatches but also due to other factors. This issue is discussed in detail in a review by Ryo Otsuka [139].

This immune rejection response can result in the destruction of transplanted cardiomyocytes, compromising their survival and integration into host tissue [128]. To improve the engraftment of pluripotent stem cell-derived cardiomyocytes (PSC-CMs) and prevent immune rejection, immunosuppressive drugs have been tested. Immunosuppressive drugs suppress the recipient’s immune response to help reduce immune reactions against transplanted cells, aiming to enhance the safety and feasibility of stem cell transplantation in regenerative medicine applications. Immune suppression using methylprednisolone, cyclosporine, and CTLA4 immunoglobulin was essential in implanting human PSC-CM in non-human primates [128].

Even with HLA-matched PSC-CMs, long-term immune inhibition is used after transplantation, even with autologous or hypoimmunogenic PSC-CMs. because rejection is not always due to HLA mismatches [139,140].

Long-term immunosuppression can lead to side effects. Therefore, alternative strategies are being explored to address immune rejection in stem cell therapies, as genetic ablation of HLA molecules from PSC combined with gene transduction of several immunoregulatory molecules may be effective in avoiding immunological rejection [140,141].

### 8.2. Scar Tissue

Fibrosis in an infarct scarred heart is a result of excess ECM production by myofibroblasts derived from fibroblasts and activated by inflammatory growth factors and cytokines, mainly (TNF)-α and (TGF)-β [141]. Therefore, converging various anti-inflammatory and immunosuppressives with tissue engineering of stem cell surfaces is essential to overcome these challenges of developing PSC-based therapy [141,142].

Several antifibrotic approaches, including pharmacological agents, gene therapies, microRNAs, modified biomaterials, biomedical engineering of PSC surface, and genetic ablation of HLA molecules, can prevent transplantation failure by optimizing the microenvironment for implantation or minimizing the inflammation at the implantation site [143,144,145].

Transient ventricular arrhythmias were observed following the injection of PSC-CMs into ischemic hearts of non-human primates and swine. The electrical coupling between the host and graft, combined with the immature nature of the PSC-CMs, may cause them to act as ectopic pacemakers [36].

### 8.3. Approaches to Deliver hPSC-CMs into the Host Heart

Human PSC-CMs have been transplanted into various animal models, including mice, rats, guinea pigs, pigs, and non-human primates. In these models, the grafted cardiomyocytes were able to survive, engraft, and provide functional benefits to injured hearts. The PSC-CMs facilitated remuscularization and formed electrical connections with the host tissue through gap junctions. Two main methods for delivering hPSC-CMs into the host heart have been identified: direct intramuscular injection of dispersed cell suspensions and epicardial attachment of EHT. Each method has its pros and cons. While EHT requires time-consuming cell manipulation and specialized skills or tools to construct the tissue, it tends to be more easily retained in the host heart tissue [36,146].

Direct injection of PSC-CMs into the heart has shown that these cells can electrically integrate with host cardiomyocytes in various animal models. However, the electrical integration of epicardial EHT with the host heart remains a subject of debate [36,133]. One of the crucial disadvantages of the direct injection of PSC-CMs has been post-transplant ventricular arrhythmia, which was detected only after cell injection in large animal studies [36,126,128,147,148].

### 8.4. Post-Transplant Arrhythmia

Post-transplant arrhythmias are considered one of the most concerning barriers to the clinical translation of CM transplantation [149,150]. Post-transplant arrhythmias have not been observed in small animal models, such as mice, rats, and guinea pigs [36,127]. However, in large animal models, transient ventricular arrhythmias were detected following PSC-CM transplantation [36,126,149]. Shiba et al. noted non-lethal ventricular arrhythmias in an allogenic transplantation model involving non-human primates [36,147]. Another study found that post-transplant ventricular arrhythmias were more frequent and potentially lethal in a porcine myocardial infarction model [36,148]. Electro-anatomical mapping identified the mechanisms of these arrhythmias in non-human primates and pigs as ectopic pacing from the graft area [36,128,148]. There are two plausible reasons why grafted PSC-CMs might act as “ectopic pacemakers” and cause post-transplant ventricular arrhythmias: their immaturity and automaticity.

### 8.5. Immaturity of PSC-CMs

According to Kadota et al. area [36], human PSC-CMs begin to beat spontaneously around 10 days after differentiation and can continue contracting for over a year in vitro. Compared to adult cardiomyocytes, they generally exhibit immature characteristics in terms of morphology, electrophysiology, and gene expression area [36,151]. Engrafted cardiomyocytes also display these immature traits before transplantation. The incidence of post-transplant ventricular arrhythmias decreases over a few weeks post-transplantation, likely due to the gradual maturation of the engrafted hPSC-CMs area [36,126,148]. Various factors, including extended culture periods area [36,152,153], chemical treatments area [36,86], physical stretching area [36,154], electrical stimulation area [36,95], and metabolic supplements area [36,87], have been shown to enhance the maturation of hPSC-CMs in vitro. Recently, combining three-dimensional culture with physical training has been reported to further advance their maturation area [36,99,155]. Although these methods do not make hPSC-CMs identical to adult cardiomyocytes, multiple stimulations are necessary to approximate adult phenotypes in vitro. Additionally, the in vivo environment can promote cell maturation after transplantation. Engrafted hPSC-CMs at the edges of the host heart tissue show more mature phenotypes compared to those in the central graft area [36,126]. Notably, hPSC-CMs engrafted in non-human primates have a larger diameter than those engrafted in rat hearts. While neonatal rat-derived CMs can mature to resemble adult CMs after transplantation in neonatal rat hearts, hPSC-CMs do not achieve the same level of maturation within the same timeframe [36,57]. The electrophysiological phenotype varies among species, affecting the in vivo maturation response between rodent-derived CMs and hPSC-CMs [127]. The immaturity of the implanted CMs is believed to cause focal automaticity, leading to arrhythmias [149,156].

### 8.6. Automaticity

In standard differentiation protocols, cardiomyocytes differentiate into ventricular, atrial, and nodal subtypes. Nodal cells make up less than 10% of the total cardiomyocytes and are characterized by a faster spontaneous beating rate and high expression of automaticity-related genes like HCN4, which is responsible for the pacemaker current. While unpurified hPSC-CMs have been shown to rescue a porcine atrioventricular block model [157], Nkx2.5-negative CMs contain a higher proportion of nodal cells [36,66,158]. Therefore, purifying pacemaker cells could facilitate the development of a biological pacemaker, whereas removing nodal cells from hPSC-CMs before transplantation might reduce the incidence of ventricular arrhythmias in larger animal models [36].

## 9. Recent Successful Trial in Primates

Most recently, Jebran et al. [18] received very promising results by using EHM allografts in a model of macaque monkeys. In macaques, EHM grafts containing 40 to 200 million cardiomyocytes/stromal cells demonstrated long-term retention (up to 6 months) and dose-dependent enhancement of the target heart wall, both in healthy hearts and those with myocardial infarction-induced heart failure. The grafts were found to enhance contractility of the target heart wall and improve ejection fraction, indicating both local and global cardiac support. Additionally, the study found no evidence of arrhythmias or tumor growth. A clinical trial using the same technique was performed in one human patient with advanced heart failure. The results confirmed successful remuscularization of the failing heart tissue without intolerable side effects such as arrhythmias or tumor formation, demonstrating the feasibility, safety, and potential efficacy of EHM therapy, marking a significant step toward broader clinical application.

## 10. Conclusions

In summary, pluripotent stem cell-derived cardiomyocytes (PSC-CMs) represent a promising avenue for the treatment of cardiovascular diseases, offering a potential solution to the limitations of conventional non-regenerative therapies and the limited availability of donor organs. Research has taken great strides in understanding the differentiation, maturation, and functional integration of PSC-CMs, supported by novel techniques such as three-dimensional cultures, metabolic modulation, mechanical and electrical conditioning, and tissue engineering. In spite of these breakthroughs, the key issues with immunological rejection, post-transplant arrhythmia concerns, inadequate cardiomyocyte development, and attaining smooth electromechanical integration with the host heart continue to be significant obstacles to practical translation. To address these issues, high-impact research directions are needed, specifically: (1) creating robust cell encapsulation strategies or hypoimmunogenic PSC lines; (2) improving maturation protocols to eliminate nodal cell automaticity and improve adult-like function; and (3) optimizing cell delivery platforms, like engineered heart muscle (EHM) allografts, to guarantee controlled electrical coupling and high retention. However, with recent advances in using EHM allografts in primates [19], there appears to be a promising way forward for human application on the horizon.

## Figures and Tables

**Figure 1 metabolites-15-00735-f001:**
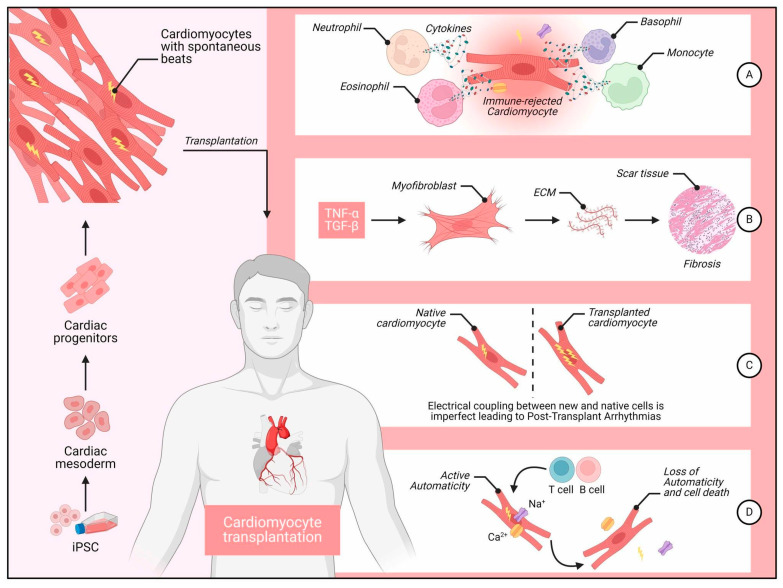
Some of the processes, challenges, and outcomes associated with iPSC cardiomyocyte transplantation. Legend: Left Panel, The derivation of cardiomyocytes from iPSCs. iPSCs are first differentiated into cardiomyocytes with spontaneous contractile activity. These cardiomyocytes are then transplanted into the patient’s heart. Right Panel (**A**) Immune Rejection: Transplanted cardiomyocytes encounter immune rejection, which involves various immune cells, including neutrophils, eosinophils, basophils, and monocytes. These immune cells release cytokines, leading to the targeted destruction of the transplanted cells labeled as “immune-rejected cardiomyocyte.” (**B**) Fibrosis Formation: After immune rejection, inflammatory cytokines like TNF-α and TGF-β promote the activation of myofibroblasts. Myofibroblasts synthesize ECM proteins, resulting in scar tissue formation and fibrosis, which impairs cardiac function. (**C**) Post-Transplant Arrhythmias: Imperfect electrical coupling between native and transplanted cardiomyocytes leads to arrhythmias. The asynchronous electrical signals between native and transplanted cells create electrical instability, causing post-transplant arrhythmias. (**D**) Loss of Automaticity and Cell Death: Some transplanted cells initially display active automaticity, generating Ca^2+^ and Na^+^ currents. However, ongoing immune response from T and B cells can cause a loss of automaticity and eventual cell death, reducing the therapeutic efficacy of the transplantation.

**Table 1 metabolites-15-00735-t001:** Overview of experimental strategies, molecular pathways, and influencing factors in PSC-derived cardiomyocyte differentiation.

Aspect	Summary	Model/System	Reference(s)
Therapeutic potential of PSCs	PSCs (ESCs and iPSCs) can generate cardiomyocytes almost limitlessly and show functional advantages after transplantation.	PSC-derived cardiomyocyte transplantation (preclinical studies)	[36]
PSCs as a model of human cardiac development	PSCs can recapitulate early embryonic heart development in vitro and generate cardiovascular cells even in poorly defined conditions (e.g., FBS).	In vitro PSC differentiation	[37]
First reports and baseline efficiency	First production of cardiac myocytes from PSCs used embryoid bodies in serum-containing media.	Embryoid bodies in serum-containing media	[38]
Enhancing differentiation via co-culture	Inclusion of mouse endodermal (END2) cells enhances differentiation efficacy.	Co-culture with mouse endodermal (END2) cells	[59]
Cytokine-driven methods (BMP4, Activin A)	BMP4 and Activin A used in 2D monolayer and EB systems efficiently generate cardiomyocytes; activinA/Nodal, Wnt, and BMP assist in primitive streak/mesoderm induction.	2D monolayer and embryoid body–based differentiation systems	[41,42,60]
Small-molecule modulation of Wnt/β-catenin and BMP	BMP-4 early treatment then small-molecule Wnt inhibitors increased cardiomyocyte production and clarified temporal Wnt activation mechanisms for consistent hiPSC-CM generation.	hiPSCs treated in vitro (BMP-4 early, then Wnt inhibitors)	[43]
MAO-A/ROS in cardiac commitment	MAO-A–mediated ROS generation is required to activate AKT and WNT signaling during cardiac lineage commitment to obtain fully functional human cardiomyocytes.	Human iPSC cardiac lineage differentiation (in vitro)	[44]
Early Wnt inhibition and cell density effects	Early Wnt production inhibitor suppressed anti-cardiac mesoderm gene expression and increased cardiomyocyte formation; cell density affects anti-cardiac mesoderm genes via auto/paracrine Wnt reduction.	Low- and high-density cell cultures (in vitro)	[45]
Nicotinamide effects	Nicotinamide increases CM formation from mesoderm progenitors while suppressing other lineages; it inhibits p38 MAP kinase and, by inhibiting ROCK, improves cardiomyocyte lifespan.	Mesoderm progenitor → CM differentiation (in vitro)	[46]
Chemically defined, serum-free protocol	A three-step differentiation system using only chemically defined factors (no serum) was presented to produce cardiac myocytes.	Chemically defined, serum-free three-step protocol (in vitro)	[47]
S-nitrosylation and maturation hypothesis	Hypothesis that increased S-nitrosylation (via NO signaling and absence of GSNOR) enhances differentiation and maturation of iPSC-derived CMs.	iPSC-derived cardiomyocytes (in vitro hypothesis)	[48]
iPSC line variability—telomeres/TRF1	iPSC^high^T (telomerase competent, high TRF1) differentiate faster/more effectively into CMs; ascorbic acid further increases CM production in iPSC^high^T; iPSC^slow^T differentiates poorly to mesoderm/endoderm but well to ectoderm.	Comparison of iPSChighT vs. iPSCslowT lines (in vitro)	[49]
BCL2 knockout effects	BCL2 KO delayed hiPSC → CM differentiation, decreased Ca^2+^ toolbox expression/activity, and reduced c-Myc expression and nuclear localization during early cardiac differentiation.	hiPSC cardiac differentiation with BCL2 KO (in vitro)	[50,51]
High cell density and minimal density studies	High cell density plating commonly used; Le et al. calculated minimal density and used a basic medium with insulin and ROCK inhibitor—after adjustments, cTnT protein produced in 10% of cells.	High-density plating (1–4 × 10^5^ cells/cm^2^) and minimal-density experiments (in vitro)	[51,52,53,54,55]
Role of auto/paracrine vs. cell–cell contact	Auto/paracrine factors, rather than direct cell–cell contact, play a significant role in cardiomyocyte development in hiPSC cocultures at both low and high densities.	hiPSC coculture (low and high densities, in vitro)	[45]
Cell death modality during early differentiation	Ferroptosis identified as the primary cell-death modality during the first 48 h of cardiac differentiation; inhibiting ferroptosis increases robustness and efficiency.	Early cardiac differentiation (first 48 h, in vitro)	[56]
In vivo maturation after transplantation	After 3 months in neonatal rat hearts, engrafted hiPSC-CMs partially matured but remained smaller than host cardiomyocytes; grafts in adult rat hearts showed greater maturation and larger grafts—host developmental stage influences maturation.	In vivo transplantation: neonatal vs. adult rat hearts (3 months post-transplant)	[57,58]

## Data Availability

No new data were created or analyzed in this study. Data sharing is not applicable to this article.

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
