# Peer review of "Recent Advances of Pluripotent Stem Cell-Derived Cardiomyocytes for Regenerative Medicine"

_metabolites, 2025, doi:10.3390/metabo15110735_

Round 1

Reviewer 1 Report

Comments and Suggestions for Authors

Major revisions required before considering publication.

General Comment: This is a well-prepared and timely review on pluripotent stem cell-derived cardiomyocytes and their role in regenerative medicine. However, revisions are needed to sharpen the critical analysis, improve structural clarity, and strengthen translational insights. By addressing these comments, the manuscript would make a valuable contribution to the field:

Major comments

  1. Scope and Focus: The manuscript sounds more descriptive than analytical; the narrative must be shifted from descriptive to analytical. For example, the differentiation protocols and maturation strategies are described in detail, but the authors could provide a stronger critical comparison of which approaches have demonstrated the most consistent translational potential. I recommend that the authors more explicitly distinguish between findings that are robust and widely replicated versus those that remain preliminary or controversial.
  2. Figures and Visuals: Additional schematic diagrams (e.g., comparing in vitro vs. in vivo maturation strategies, or delivery methods) would enhance readability.
  3. Clinical translation Section: The clinical implications in immune rejection, arrhythmia risk, and fibrosis seem under-discussed. The authors should provide a critical assessment of whether current strategies (e.g., hypoimmunogenic PSC lines, genetic editing) are realistic in the short-to-medium term for clinical application.
  4. Balance of Content: While the manuscript excellently and in detail describes overview of differentiation and maturation methods, the section on preclinical and translational applications could be expanded. For example, large animal studies are critical to clinical translation and are very briefly described; they should be discussed in more depth.
  5. Novelty: A more explicit framework that categorizes strategies into "promising for near-term clinical use" versus "experimental with long-term potential" would increase the utility of the paper for the readership. At present, the manuscript is overly descriptive and demands more structure.

Minor Comments:

  • Condense dense protocol descriptions or present them in tables for improved clarity.
  • Perform a careful language edit to correct grammar, style, and consistency.
  • Ensure all abbreviations are clearly defined at first mention in the text.
  • Add recent clinical trial registries or society position papers to strengthen the references.
  • Revise the conclusion to highlight the most critical barriers and high-impact future directions.

Author Response

We thank the reviewer for their very helpful comments. Before are our responses:

Comment1: Scope and Focus: The manuscript sounds more descriptive than analytical; the narrative must be shifted from descriptive to analytical. For example, the differentiation protocols and maturation strategies are described in detail, but the authors could provide a stronger critical comparison of which approaches have demonstrated the most consistent translational potential. I recommend that the authors more explicitly distinguish between findings that are robust and widely replicated versus those that remain preliminary or controversial.

Response: The aim of this narrative review is to help researchers establish a theoretical framework and focus or context for their research. Our aim was to identify patterns and trends in the literature so that researchers can identify gaps of knowledge in the field. Our aim is to provide a descriptive summary that can serve as a starting point for future research.

Comment 2: Figures and Visuals: Additional schematic diagrams (e.g., comparing in vitro vs. in vivo maturation strategies, or delivery methods) would enhance readability.

Response: We added two tables to the manuscript, one provides an overview of the experimental strategies, molecular pathways, and influencing factors in PSC-derived cardiomyocyte differentiation and another summarizing factors enhancing structural, electrophysiological, and metabolic maturation of pluripotent stem cell–derived cardiomyocytes.

Comment 3: Clinical translation Section: The clinical implications in immune rejection, arrhythmia risk, and fibrosis seem under-discussed. The authors should provide a critical assessment of whether current strategies (e.g., hypoimmunogenic PSC lines, genetic editing) are realistic in the short-to-medium term for clinical application. Balance of Content: While the manuscript excellently and in detail describes overview of differentiation and maturation methods, the section on preclinical and translational applications could be expanded. For example, large animal studies are critical to clinical translation and are very briefly described; they should be discussed in more depth. Novelty: A more explicit framework that categorizes strategies into "promising for near-term clinical use" versus "experimental with long-term potential" would increase the utility of the paper for the readership. At present, the manuscript is overly descriptive and demands more structure.

Response: These are great points. Unfortunately, since our review is already 26 pages long expanding the discussion on the translational potential of these studies would require an additional 25% increase in the length of the manuscript. However, we are currently exploring the idea of writing a whole separate review of this topic, focusing solely on the translational potential of the different models.

Comment 4: Condense dense protocol descriptions or present them in tables for improved clarity.

Response: We added two tables to condense dense protocols.

Comment 5: Perform a careful language edit to correct grammar, style, and consistency.

Response: We have conducted a careful language edit to correct grammar and style.

Comment 6: Ensure all abbreviations are clearly defined at first mention in the text.

Response: We have reviewed this issue and made sure that all abbreviations are clearly defined at first mention in the text.

Comment 7: Revise the conclusion to highlight the most critical barriers and high-impact future directions.

Response: The conclusion has been rewritten to address this comment.

Reviewer 2 Report

Comments and Suggestions for Authors

Review Report

Reviewer 1 Comments:

The manuscript addresses an important and timely topic in regenerative medicine, particularly given the global burden of heart failure and the limitations of current therapies such as heart transplantation. The work has potential to serve as a solid review; however, it requires substantial revision to align with the scope of Metabolites. Specifically, the integration of metabolomics and metabolic regulation into key sections is necessary. If this metabolic perspective cannot be convincingly incorporated, resubmission to a more suitable journal should be considered.

Major Comments

Abstract

Improve transitions to provide logical flow. For example, clearly articulate how developmental insights inform protocols, linking pathology to PSC methods.

Methodology Section

A dedicated methodology section outlining how the review was conducted should be added.

Section 2 should explicitly link the different types of cardiomyopathies to PSC opportunities under both preclinical and clinical contexts.

The review is largely descriptive, cataloging differentiation protocols (e.g., BMP4/Activin A, Wnt modulation) without critical assessment.

Provide comparative discussion: for instance, early embryoid body methods yielded 5–10% cardiomyocytes, whereas modern small-molecule protocols achieve 80–90%.

A quantitative summary table reporting yields, purity, and maturation markers would significantly strengthen the manuscript.

There is no discussion of metabolic shifts in PSC-derived cardiomyocytes (e.g., transition to fatty acid oxidation, mitochondrial maturation).

Discuss how metabolomic profiling can be leveraged to optimize differentiation and maturation strategies.

Line 230: Please provide appropriate references for several cited studies.

Line 648: Indicate references for the various animal models discussed.

Expand the discussion on implications for therapeutic use of iPSCs, integrating both preclinical and clinical perspectives.

At 26 pages, the manuscript is overly lengthy for a review and risks losing reader focus. Consider condensing content while maintaining critical analysis.

Author Response

We thank the reviewer for their detailed comments and we addressed the majority of comments as follows:

Comment 1: Improve transitions to provide logical flow. For example, clearly articulate how developmental insights inform protocols, linking pathology to PSC methods.

Response: We rewrote the abstract to improve the logical flow, as requested.

Comment 2: A dedicated methodology section outlining how the review was conducted should be added.

Response: This work is a narrative literature review, not a systematic review. Therefore, a methodology section is not often included nor is it required.

Comment 3: Section 2 should explicitly link the different types of cardiomyopathies to PSC opportunities under both preclinical and clinical contexts.

Response: We have added three paragraphs to explicitly link the different types of cardiomyopathies to PSC opportunities, in section 2.

Comment 4: The review is largely descriptive, cataloging differentiation protocols (e.g., BMP4/Activin A, Wnt modulation) without critical assessment.

Response: The aim of this narrative review is to help researchers establish a theoretical framework and focus or context for their research. Our aim was to identify patterns and trends in the literature so that researchers can identify gaps of knowledge in the field. Our aim is not to judge or criticize what has already been published, but rather to provide a descriptive summary that can serve as a starting point for future research.

Comment 5: Provide comparative discussion: for instance, early embryoid body methods yielded 5–10% cardiomyocytes, whereas modern small-molecule protocols achieve 80–90%. A quantitative summary table reporting yields, purity, and maturation markers would significantly strengthen the manuscript.

Response: While we have not included a qunatitative comparative discussion, we opted instead to provide two tables one comparing the experimental strategies, molecular pathways, and influencing factors in different models, and another summarizing factors enhancing structural, electrophysiological, and metabolic maturation.

Comment 6: There is no discussion of metabolic shifts in PSC-derived cardiomyocytes (e.g., transition to fatty acid oxidation, mitochondrial maturation). Discuss how metabolomic profiling can be leveraged to optimize differentiation and maturation strategies.

Response: These are all great points. But unfortunately this is a very significant body of work that might require a separate review article all by itself. If included, it would almost double the size of our (already overly lengthy) review, and it is thus beyond the scope of our work here.

Comment 7: Line 230: Please provide appropriate references for several cited studies. Line 648: Indicate references for the various animal models discussed.

Response: We added the appropriate references missing.

Comment 8: Expand the discussion on implications for therapeutic use of iPSCs, integrating both preclinical and clinical perspectives. At 26 pages, the manuscript is overly lengthy for a review and risks losing reader focus. Consider condensing content while maintaining critical analysis

Response: Expanding the discussion would make the review even longer, however we tried to rewrite the conclusion section to address some of these concerns.

Round 2

Reviewer 2 Report

Comments and Suggestions for Authors

The article has been revised and considered for publication